# The dynamics of İzmir Bay under the effects of wind and thermohaline forces

Erdem Sayın[1] and Canan Eronat[1]

[1]Institute of Marine Sciences and Technology, Dokuz Eylül University, İzmir, Turkey

5   *Correspondence to*: Erdem Sayın (erdem.sayin@deu.edu.tr)

**Abstract.** The dominant circulation pattern of the İzmir Bay by the Aegean Sea coast of Turkey is studied under the influence of wind and thermohaline forces. İzmir Bay is discussed by subdividing the Bay into outer, middle and inner areas. Wind is the most important driving-force in İzmir coastal area. There are also thermohaline forces due to the existence of distinguished water types of different physical properties in the Bay. Instead two-layer stratification during summer, a homogeneous water column exists in winter. The free surface version of the Princeton model (Killworth's 3D general circulation model) is applied with the input data obtained through the measurements made by the research vessel K. Piri Reis. As a result of the simulations with artificial wind, the strong consistent winds generate circulation patterns independent from the seasonal stratification in the Bay. Wind-driven circulation shows cyclonic or anti-cyclonic movements in the middle bay where the distinguished İzmir Bay Water (IBW) forms. Cyclonic movement takes place under the influence of southerly and westerly winds. On the other hand, northerly and easterly winds cause an anti-cyclonic movement in the Middle Bay.

**Key words:** Circulation, 3D numerical model, İzmir Bay

## 1 Introduction

İzmir Bay, located on the Aegean Sea coast of Turkey, was formed as a part of the Aegean Sea during the Pleistocene period. Later, it was partially filled with the silt carried by Gediz River (Maddy et al., 2012). The Bay can be divided into three areas according to their physical characteristics (containing different water types and bathymetry, etc.): Outer, Middle and Inner Bays, as indicated in Fig. 1. The average depth in the Outer Bay is about 70 m. It decreases significantly towards the Inner Bay to an approximate depth of 10 m.

In the Bay, three distinguished water types exist: Aegean Sea Water (ASW), İzmir Bay Water (IBW) and İzmir Bay Inner Water (IBIW). Outer Bay water type ASW has a greater volume than the other water types in the bay. Relatively small temporal changes are observed in its temperature and salinity values due to its large volume. Inner Bay water type IBIW is the coldest in winter and its temperature varies from 9.1 °C to 13.9 °C. It has maximum temperature in summer and changes from 24.6 °C to 27.5 °C. IBW is formed in the middle Gyre area, influenced by the Gediz River inflow and by the upwelling and downwelling processes that are mainly driven by southerly and northerly winds, respectively. IBW seems

that it is a mixture of IBIW and ASW in winter. But it is very distinguished water type in summer with its
higher salinity values varying between 39.6 psu and 39.9 psu (Sayın et al., 2006).

The water masses of shallow bays as İzmir Bay have impacts on the Aegean Sea through temperature or salt controlled dense water cascading. For example, the dense IBW, which is always saltier than ASW in summer and denser in winter, flows near the coast of Foça over the topography into the Aegean Sea. (Sayın et al., 2006; Eronat & Sayın, 2014).

The water exchange through some vertical sections can give a good estimate about renewal time for the selected regions in the İzmir Bay. The renewal time, which is important from the biological and chemical point of view, considering the water exchange through the vertical section between İzmir Bay and Aegean Sea is found by 46 and 29 days for the winter and summer case, respectively (Sayın, 2003). The water input through Gediz River is relatively low (averaging about 40.3 $m^3$/s) over 30 years compared to the

water exchange with the Aegean Sea (approximately 7000 $m^3$/s). Since Gediz River used to discharge into the Inner and Middle Bays until 1890, there is an important accumulation of sedimentation in the Inner and Middle Bays. This situation, causing major differences in the bathymetric structure of the Inner Bay, has a very important role on the circulation there and avoids massive water exchange between the Inner Bay and the Middle Bay. The effect of shallowing caused by the previous Gediz river bed on the

water motion in the inner bay section of Izmir Bay is investigated by Karahan (2002) and the results were discussed for various meteorological conditions. Using a three dimensional finite difference model the author demonstrated that the shallowing which is in the middle bay entrance has effect on the water input and output of outer bay.

Suspended material and food can be trapped in the interfacial area between the water masses from the

Aegean Sea and the interior parts of the Bay. Therefore, these places are the attraction locations for fish. Depending on the wind condition and stratification in the water column, the current from the Karaburun area carries fish eggs and larvae origin from Mediterranean into the small Gülbahçe Bay through the Mordoğan Passage and even further to the Middle Bay. This feature adds to the diversity of marine life, as can be observed from various larvae found in Gülbahçe Bay (Sayın and Öztürk, 2006).

The analysis of current system in the Bay is quite recent: The first mathematical model study related to the circulation pattern of the İzmir Bay is the depth-averaged two dimensional mathematical model given in Karahan (1988). The current system in the Bay has been examined by Saner (1994). He has calculated the circulation pattern and water exchange between the different regions of the İzmir Bay using two and three dimensional wind-driven mathematical models. Saner (2005) has also compared his two models:

The two dimensional model solving the equations using standard 2D-ADI method with the 3-D dimensional sigma coordinate model. Pazı (2000) has studied the current system of the Bay mainly related to the observations and has found that the current is driven by wind and also by thermohaline forces. Sayın (2003) has investigated the important physical features based on observations and modelling studies. Sayin et al (2006) has run the model with realistic forcing. In this study, our motivation is to

understand the behaviour of the current field under blowing strong wind from four main directions. If the wind continues long time, approximately more than 12 hours, the current fields in the Bay will go under the influence of wind and form wind-driven circulation. There is a demand by other disciplines such as

biologists chemists etc. to get detailed information about dominant circulation patterns under characteristic wind fields.

Beşiktepe et al. (2011) calculated the circulation pattern of the İzmir Bay with a primitive equation model of the Harvard Ocean Prediction System. They have identified the elements of the sub-basin scale circulation and found that the size, structure and evolution of the main model circulation and gyres are in agreement with observations. In their simulations the mean circulation of the Bay is cyclonic gyre which occupies almost whole basin. This circulation is driven and modified by the wind and offshore forcing.

Eronat (2011) and Eronat & Sayın (2014) studied on the temporal evolution of the water characteristics of Izmir Bay including the EMT period giving information about the major deep-water formation episode and after in the Aegean Sea.

In the present study, emphasis is given to the dynamics of recirculation maintaining in İzmir Bay. These circulation patterns are formed under special wind conditions depending on background stratification. The

seasonal regime of current system and forming of persistent or quasi-permanent water movements in the Bay are relevant to studies on the biological or chemical oceanography of the Bay.

This study can also be an example for the other bays along the coastal area of the eastern Aegean Sea. The bays play crucial roles in the forming water masses in the Aegean Sea according to their water exchange potentials. A more solid foundation to our knowledge of the circulation and hydrography of the

İzmir Bay is therefore of considerably wider interest than merely regional.

Materials and methods of this study are explained briefly in the second section. The third section examines the recirculation patterns that are frequently observed in the İzmir Bay environment. The last section is the conclusion.

## 2 Material and methods

The physical oceanographic measurements have been initiated in 1988 in İzmir Bay. Since 1996 regular measurements have been conducted with Seabird CTD (Conductivity, Temperature and Depth) system.

A z-level Killworth's 3-D general circulation model based on the primitive equations described by Bryan (1969) and Cox (1984) is applied to the İzmir Bay. The specific model configuration used here is an explicit free surface version of the Princeton model, developed by Killworth et al. (1989). The Killworth

model filters the fast oscillations, letting geostrophic balance remain behind after the establishment of the steady current. The steady current is achieved by controlling the kinetic energy of the system. The integration is stopped as soon as the kinetic energy level reaches a plateau. To set a realistic stratification, selected winter and summer hydrological cruise CTD data is prescribed in the model as an initial condition. The winter and summer initial temperature and salinity values are shown in Fig. 2. The

simulations were used to define the general circulation patterns of the Bay by using real topography.

The model has been using for a long time in our institute. The validation with observations has been carried out and first results have been obtained by Sayın (2003) and Sayin et al. (2006). The comparison with the other models in our institute (a primitive equation model of the Harvard Ocean Model and FVCOM (the Unstructured Grid Finite Volume Community Ocean Model)) are done and achieved good

agreement. The model was previously applied to the Baltic Sea and the straits between Baltic and North Sea (Sayin and Krauss, 1996) justifying that this model can be used also for small seas as well as straits and channels. Sayin et al. (2006) has run the model with realistic forcing for the İzmir Bay.

In this study, the chosen model parameters are given in Table 1. No tidal and heat forcing is included in the model. The density-driven experiment is conducted using the temperature and salinity fields (no wind)
to understand the effect of the stratification on the circulation pattern. Fig. 3 shows the wind direction and average wind intensity from 2000 up to now monthly and yearly for İzmir Bay environment. The wind from north is predominant wind direction for the İzmir Bay. The average wind speed is about 5 m/s.

In this study it is tried to understand the behaviour of current field under blowing consistent strong wind. Therefore, the wind intensity is increased from 0 to 10 m/s in the model experiments testing which wind
intensity enough to simulate the strong wind condition. Then, the wind-driven experiment is conducted using wind from four main directions including the direction of predominant wind with constant intensity (5 m/s) to simulate the persistent wind condition.

The model domain is connected to the Aegean Sea. Observed temperature and salinity values are prescribed at the boundary and relaxed during rest model integration time. At the boundary, Stevens
(1990) active open boundary condition for the tracer field, Orlanski (1976) radiation condition for the external mode was applied. Stevens active boundary was chosen to force the model with observed temperature and salinity values at the boundary. It was deemed suitable to choose radiation condition for the barotropic part due to lack of consistent surface elevation information related to the Aegean Sea general circulation dynamics.

The numerical experiments can be summarized into two groups: One group focuses on the thermohaline circulation (thermohaline circulation is defined as the circulation evolved under the influence of the density-induced forces generated as a result of background temperature and salinity stratifications) and the other group with the wind-driven circulation. The model integration is variable for every run. But it takes approximately three days. The background stratification remains not changed from its original
prescribed form because of the equilibrium is succeeded in a short time. This is the importance of wind-driven scenarios with constant wind intensity. Model will not be in a steady state if we use actual wind field to run the model.

It is certain that the open sea flow is very influential for the existence of features in the real sea. In our case, we have no sea level measurements. It is not possible to get reliable sea level information from
satellite data (TOPEX/POSEIDON) due to its coarseness. On the other hand, although the İzmir Bay has a link with the Aegean Sea, cape and islands off the northern part of the Bay probably form a barrier for the development of a big sea level gradient extending from north up to İzmir Bay mouth opening to the Aegean Sea.

## 3 Results and discussions


The representation of recirculation patterns that are formed by existence of winter and summer stratification in the water column are given as a result of thermohaline circulation in Fig. 4.

First we describe the water types related to the stratification in the Bay. The existing water types in the Inner, Middle and Outer Bay evolving under the different physical processes and mechanism influence the water circulation in the Bay (Sayın et al., 2006). From the water types, the İzmir Bay Water (IBW) is denser than the Aegean Sea Water (ASW) in winter and the expected thermohaline circulation is cyclonic along the basin width (Fig. 4 upper panel). It means dense water of IBW tends to flow along the east coast towards Aegean Sea and less dense ASW that enters near Karaburun flows through the Mordoğan Passage into the Middle Bay.



The Outer Bay water (ASW) is always denser than the Middle and Inner Bay waters because of having lower temperatures near Aegean Sea in summer. Therefore, the flow from Aegean Sea is towards south and follows the path in the middle of the Outer Bay and turns towards Gediz River and then flows near the east coast of the Uzunada Island and bifurcates (Fig. 4). One branch combines with the current at the west coast of İzmir Bay and other branch combines with the strong current at the east coast of İzmir Bay making a big cyclonic circulation in the middle area. The coastal currents at both sides flow towards north compensating the flow from the Aegean Sea. The current at the east coast leaves the İzmir Bay near the coast of Foça. The summer circulation pattern is more complicated compared to the pattern obtained for winter circulation. Vertically stratified water column changes the behaviour of the current during summer. Contrary, Outer Bay water (ASW) is less dense than the Middle and Inner Bay waters in winter because of having higher temperature. Comparison winter patterns with the patterns formed in summer indicate that, except the cyclonic circulation patterns **A** (Aegean Recirculation Pattern) and **O** (Outer Bay Recirculation Pattern) in the outer Bay, the pattern **M** (Middle Bay Recirculation Pattern) changes sign. Thermohaline circulation **I** (Inner Bay Recirculation Pattern) is very weak in the Inner Bay in both seasons with a dipole shape that changes signs from winter to summer (Fig. 4).



After analysis of thermohaline circulations, some model experiments are conducted using the winds from four main directions: westerly, easterly, northerly and southerly. Persistent westerly wind changes thermohaline cyclonic circulation and winter homogeneous water is immediately under the influence of wind force (Fig. 5). Coastal jets are produced along both coasts in the wind direction and a slow return flow compensates this transport in the central area of the basin as explained in the literature. In the case of a westerly wind, as expected, characteristic flows near both coasts are in the wind direction. The recirculation pattern **M** is anticyclonic due to establishing a stronger current near the east coast of the Bay relative to the coastal current near the east coast of Uzunada Island. This pattern **M** is also anticyclonic in summer, but it is not well developed. It means that the stratification can play an important role on the current system.



In the Inner Bay, recirculation dipole pattern **I** is observed both in summer and in winter with a same sign under the west wind condition. There is no significant communication between Inner and Middle Bays because of the existence of a narrow passage (Yenikale passage). The numerical experiment was


conducted to show the development of circulation in the Inner Bay by increasing the wind intensity from zero to 5 m/s. The current, not only in the Inner Bay, but also in the other regions of the Bay starts to set up after a certain wind speed is exceeded. The current is very weak in the Inner Bay without the existence of wind force. The currents get stronger with increasing wind speed. Recirculation patterns which exist in the Middle Bay become well-developed after the increase of wind intensity above approximately 2.5 m/s and are observable both in the barotropic and baroclinic fields.

In a similar manner, the current pattern of the Bay can be analyzed for the other wind conditions as was already done for the westerly wind (Fig. 5, Fig. 6, Fig. 7 and Fig. 8). The current system is explained giving emphasis only to the recirculation patterns forming in the Bay. For example, closed circulation pattern **M** has an anti-cyclonic character under westerly and northerly wind conditions (Fig. 5 and Fig. 7); and has cyclonic character under other wind conditions; easterly and southerly (Fig. 6 and Fig. 8) for both seasons. Because of existing strong stratification in summer, a dipole forms in the middle area instead of one anti-cyclonic circulation (Fig. 5 and Fig. 7).

Circulation pattern **I** generally has dipole character in westerly and easterly wind conditions (Fig. 5 and Fig. 6). These poles change places (signs) with each other depending on changing wind directions. It has anti-cyclonic character in southerly wind condition and cyclonic character in northerly wind condition for both seasons (Fig. 7 and Fig. 8).

Circulation pattern **O** forms mainly cyclonically in case of westerly and northerly wind conditions (Fig. 5 and Fig. 7) and anti-cyclonically in case of easterly and southerly winds in both seasons (Fig. 6 and Fig. 8).

Circulation pattern **A** is not persistent. If it forms, it will be in cyclonic form. This cyclonic behaviour is seen both in summer and winter thermohaline circulations.

The small pattern formed above the **M**, most of time has the same sign as **M**. This circulation is formed at first in the Middle Bay area and later it moves towards the north. It sometimes combines with the patterns in the Outer Bay forming one big recirculation pattern. If the sign of **M** and **O** are same and **M** is very near to Outer Bay, these features combine each other.

The results obtained by the numerical method can be summarized schematically with respect to frequently seen recirculation patterns (Fig.9). İzmir Bay is very sensitive to wind intensity and direction. A central recirculation pattern **M** is frequently present in the Middle Bay. Its direction of vorticity depends strongly on the recently blowing wind condition. The other recirculation patterns, **A**, **O**, and **I**, are quasi-stationary. For example, the recirculation patterns seen in Outer Bay near the Aegean Sea **A** and **O** are developed mainly in the Middle Bay area. Sometimes they combine with the features in middle area forming a larger cyclonic or anti-cyclonic pattern in Outer Bay. They are mainly related to strong currents maintained near coastal shallow areas and gain velocity-shear related to topography. The velocities are stronger in shallower areas compared to the velocities in relatively deeper parts. Recirculation Pattern **I** is observed in the Inner Bay generally with dipole shape.

## 4 Conclusions


The circulation and water movements of İzmir Bay can be summarized as follows:

The expected basin-wide circulation in İzmir Bay is cyclonic. The Izmir Bay Water flows along the east coast towards Aegean Sea while the Aegean Sea Water enters through the Mordoğan Passage into the Bay in winter. However, in summer, although the circulation is cyclonic again, Aegean Sea Water flows into the Middle Bay near the east coast of the Uzunada Island.


**A** (Aegean Recirculation Pattern) and **O** (Outer Bay Recirculation Pattern) form in the Outer Bay. The most often observed **M** (Middle Bay Recirculation Pattern) forms in the Middle Bay and **I** (Inner Bay Recirculation Pattern) is observed in the Inner Bay generally with dipole shape.

Pattern **M** has a cyclonic character in case of southerly and easterly winds and has an anticyclonic character in case northerly and westerly winds. Sometimes in summer, anticylonic circulations cannot be developed well because of the strong background thermohaline cyclonic circulation. The recirculation pattern formed above the **M** towards the Aegean Sea has always the same sign as **M**. This pattern forms at first in the Middle Bay area and it moves to the north. It sometimes combines with the other pattern in the Middle Bay remaining one big pattern behind. This shows us that the Middle Bay area plays important role in the generation of closed recirculation patterns in the Bay.



The cyclonic middle gyre **M** is important for İzmir Bay environment from two points. First is related to the dense water formation. The densest water (IBW) forms in the Middle Bay as a result of winter convection enhanced with cyclonic circulation in winter season. It causes a dense water cascading from İzmir Bay to Aegean Sea. The latter is important from the biological point of view, forming upwelling brings nutrients rich water to the surface.


Outer Bay Recirculation Pattern **O** forms mainly anti-cyclonically in case of southerly and easterly winds and cyclonically in case of westerly and northerly wind conditions in both seasons.

The wind-driven recirculation pattern **I** almost has double poles in easterly and westerly wind conditions. These poles change places with each other depending on the wind direction. The dipole character of circulation gains cyclonic or anti-cyclonic behaviour, in turns, in northerly and southerly wind condition.


Circulation pattern **A** is not persistent. Its shape changes depending on the Aegean Sea boundary condition.

İzmir Bay research needs a synthesis including all disciplines related. In this direction, more modelling efforts besides the wind-driven circulations are necessary. The prognostic modelling approach can be a future challenge for the modelling of İzmir Bay with adding more meteorological information inside.


**Acknowledgements**

The work was carried out in the framework of the Izmir Bay Marine Research Project. We acknowledge IMST/DEU for supporting the cruises. We also extend our thanks to the people participated in the cruises. Special thanks are to Prof. Dr. Deniz ÜNSALAN for his help with improving the manuscript.

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

## Tables

| Parameters | |
|---|---|
| Horizontal resolution: | 500 m |
| Number of vertical layers: | 6 |
| Layer thickness (m) (layer 1,2,3,4,5 and 6) | 5, 10, 15, 15, 15 and 10 m |
| Horizontal eddy coeff for momentum: | 1.0E5 cm$^2$/sec |
| Vertical eddy coeff for momentum: | 1.0 cm$^2$/sec |
| Horizontal eddy coeff for heat: | 1.0E5 cm$^2$/sec |
| Vertical eddy coeff for heat : | 0.1 cm$^2$/sec |
| Baroclinic time step: | 200 sec |
| Barotropic time step: | 5 sec |
| Bottom drag coefficient | 2.2E-3 |

**Table 1**. The chosen model parameters for the wind-driven circulation experiments.

**Figures**

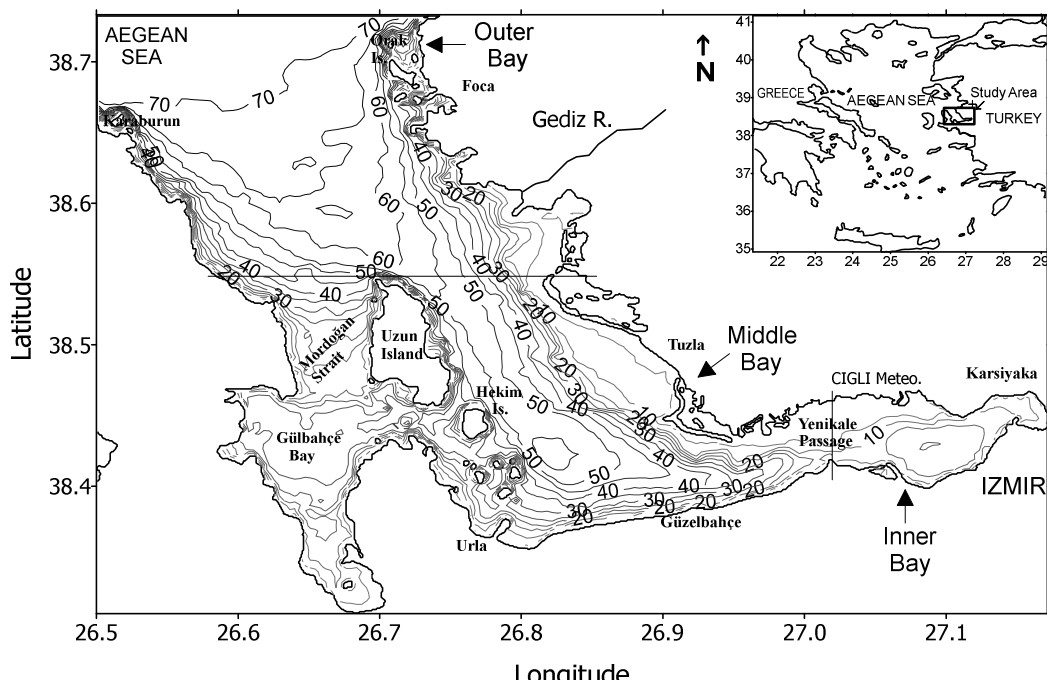

**Fig. 1.** Location of İzmir Bay and study area bathymetry.


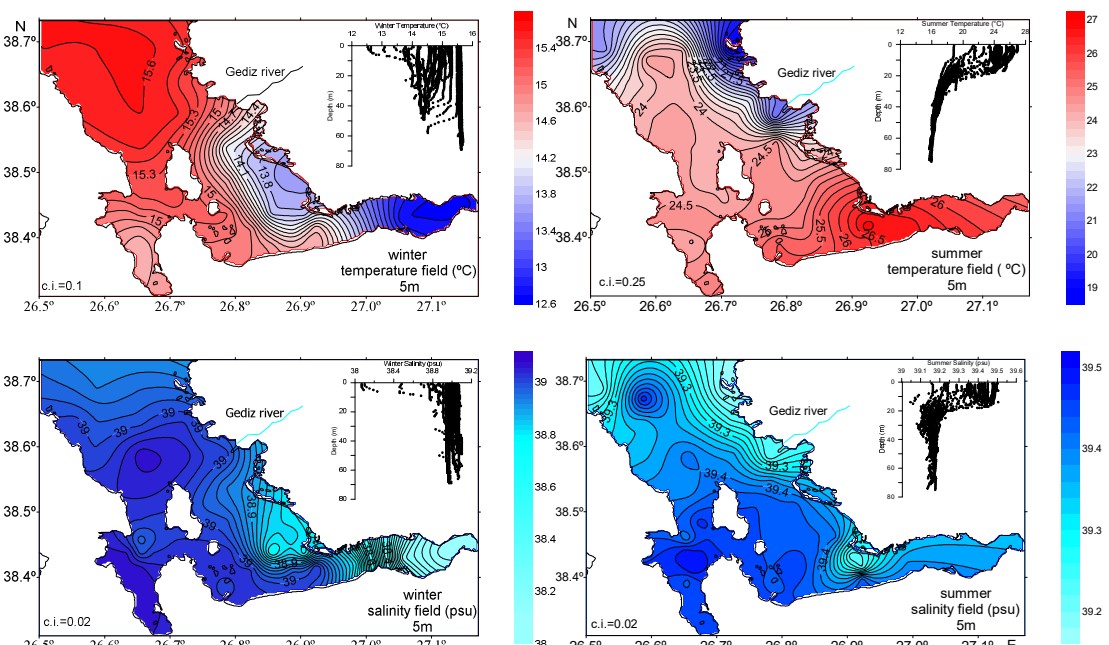

**Fig. 2.** The winter and summer, temperature and salinity fields are prepared to give to the model as temperature and salinity distribution of the first level. The related profiles are at the right-upper corner of the figures.


**Fig. 3.** Monthly and yearly wind direction and average wind intensity from 2000 up to now for İzmir Bay environment. The arrows point in the direction that the wind is blowing. The last column indicates the dominant wind and its intensity (https://www.windfinder.com).

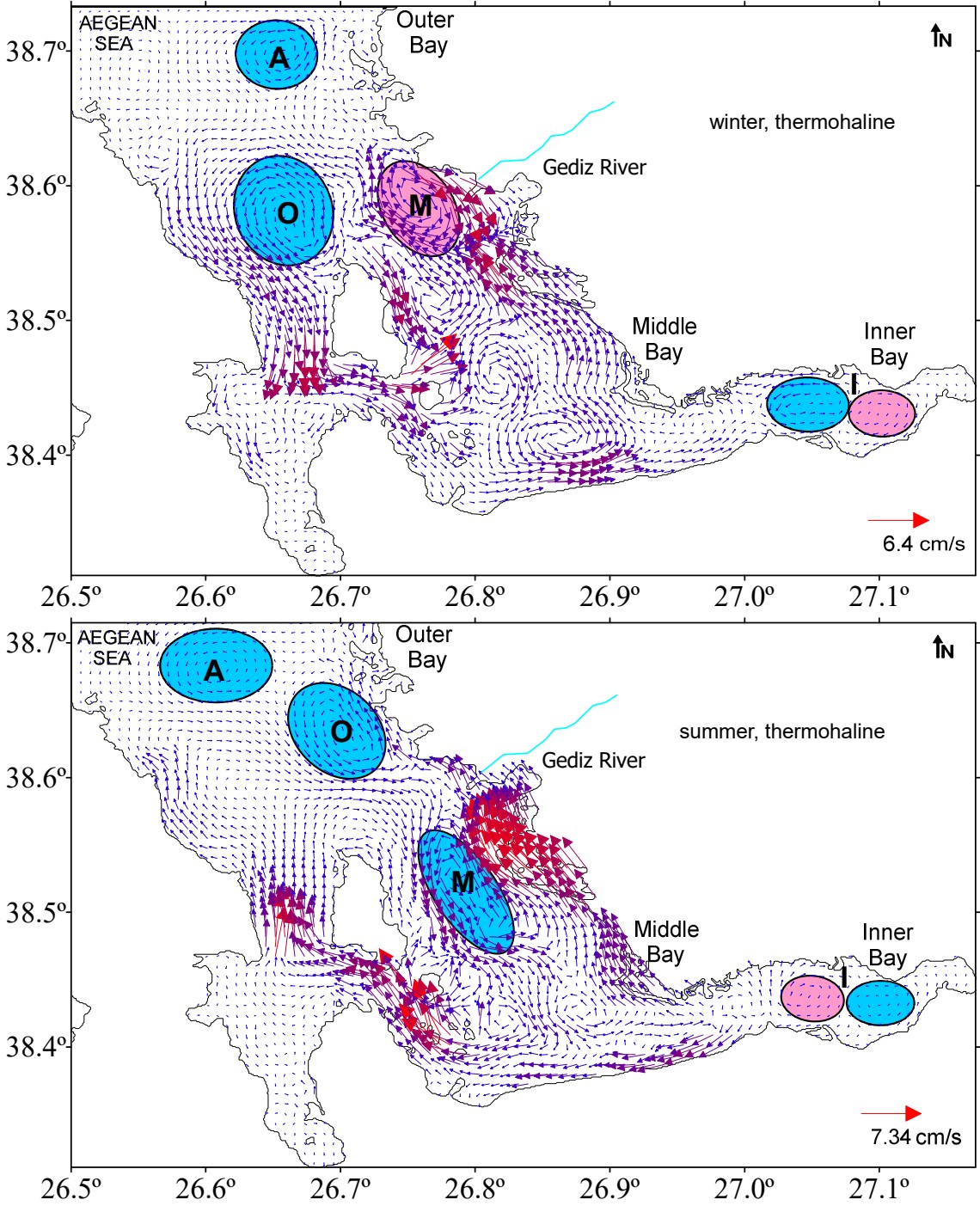

**Fig. 4.** The winter and summer thermohaline circulation (depth averaged) in the Bay.

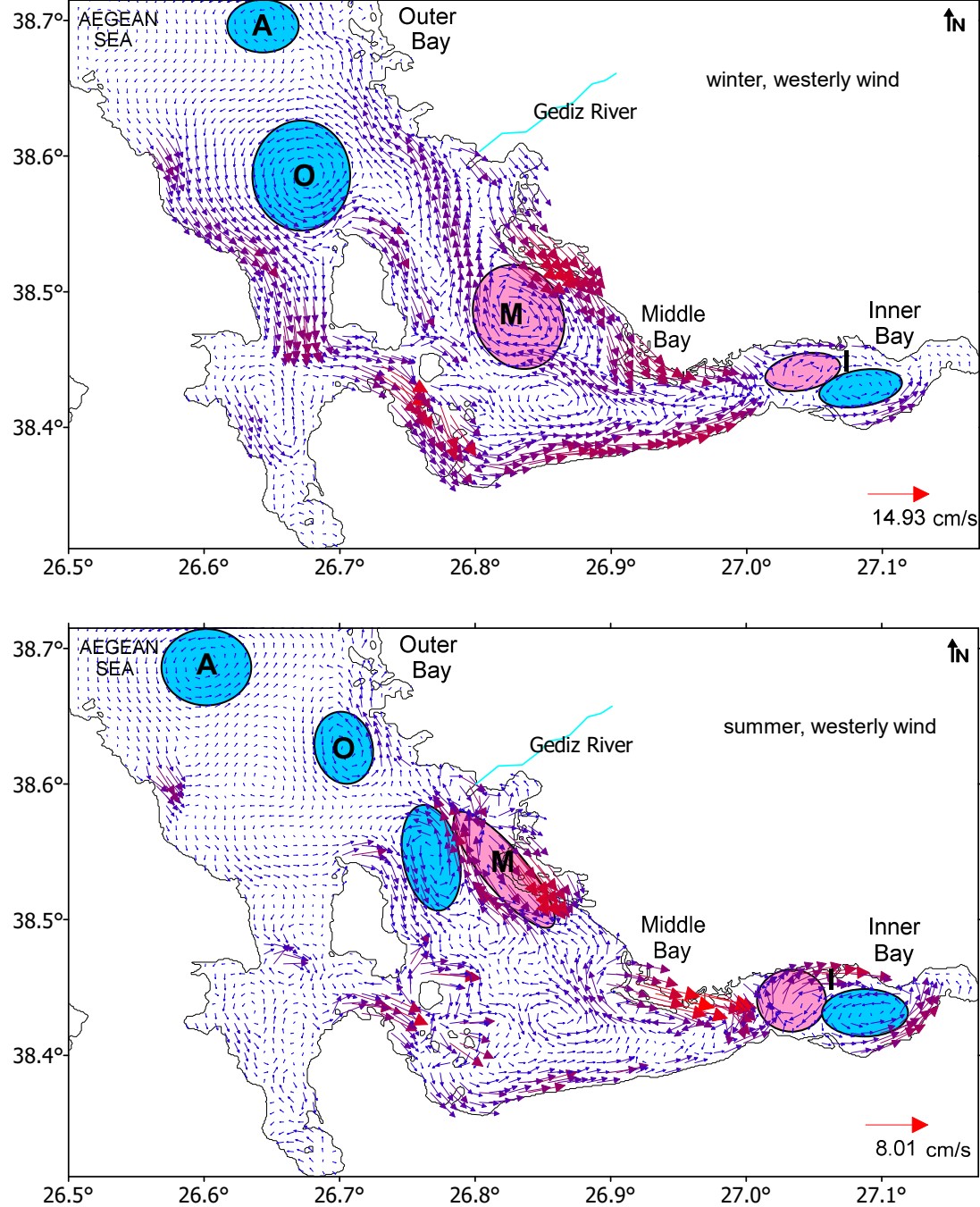

**Fig. 5.** The barotropic current pattern (depth averaged) in case of westerly wind in winter and in summer.

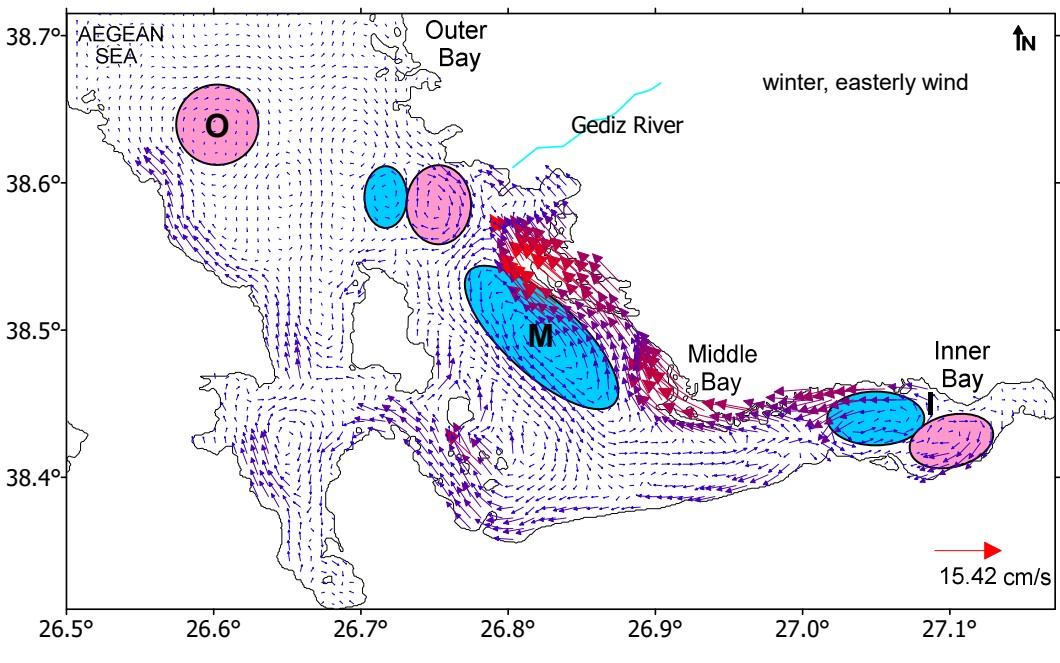

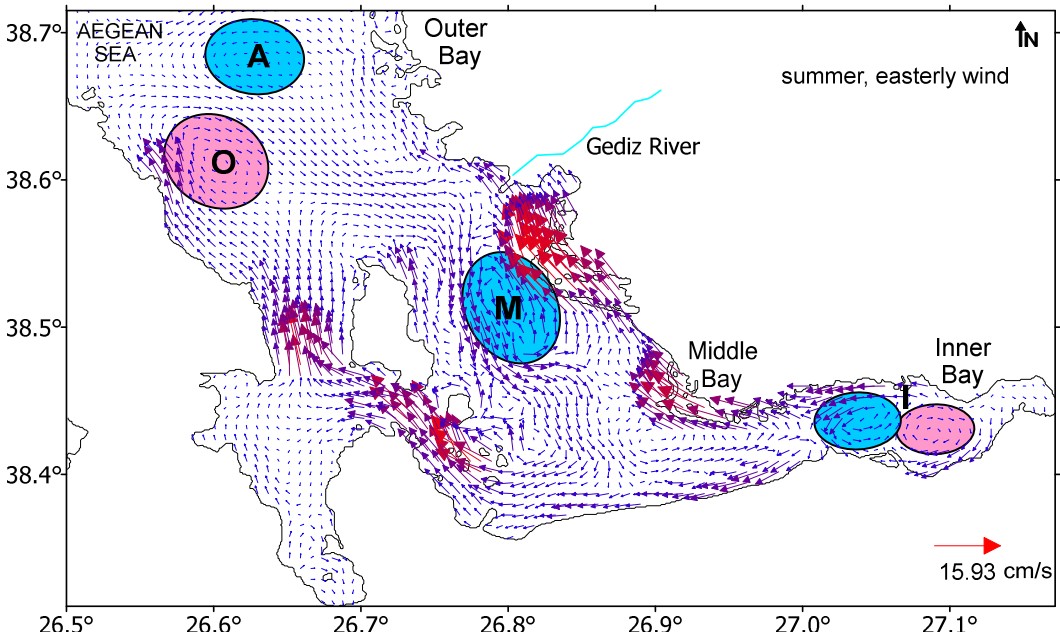

**Fig. 6.** The barotropic current pattern (depth averaged) in case of easterly wind in winter and in summer.

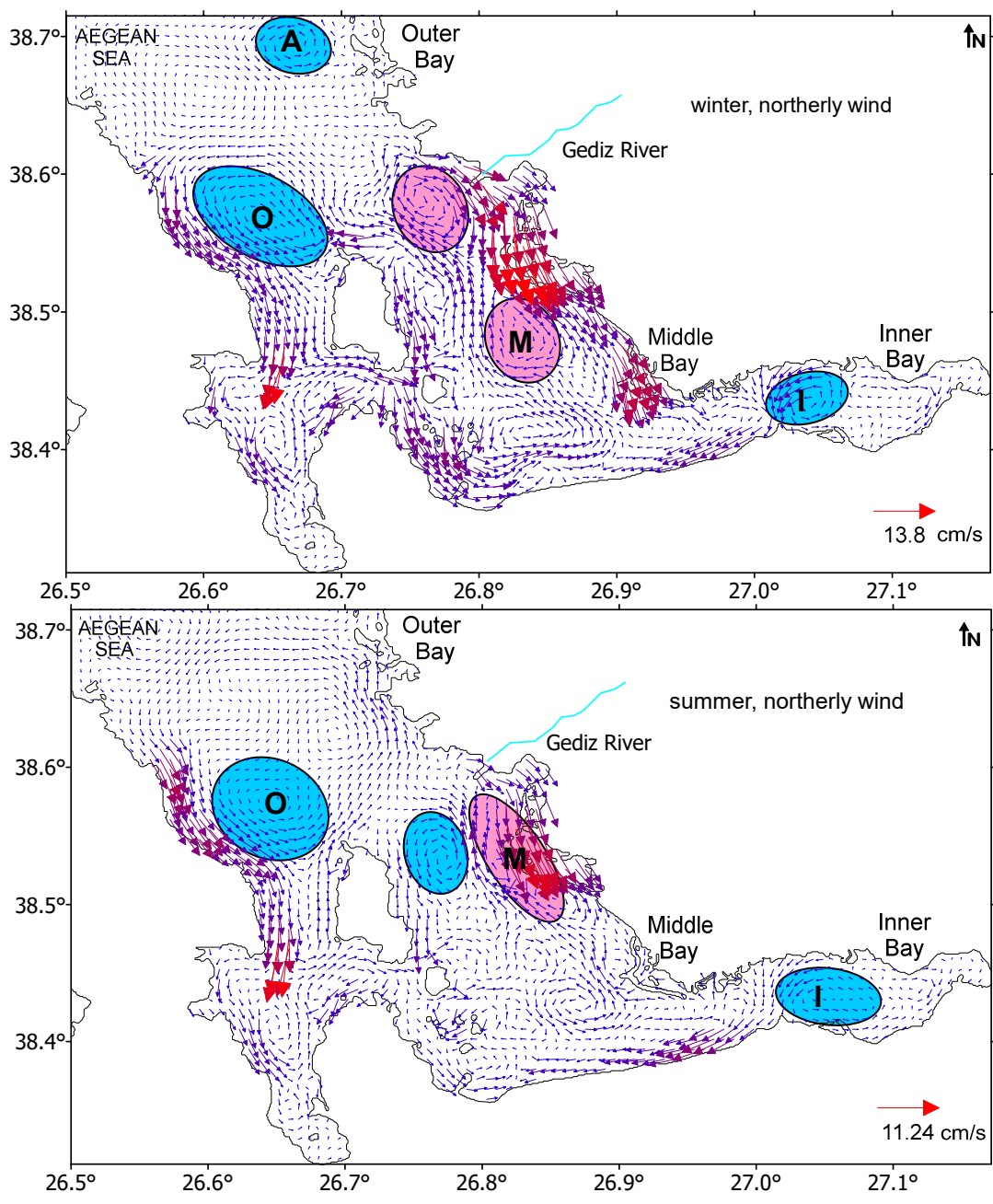

**Fig. 7.** The barotropic current pattern (depth averaged) in case of northerly wind in winter and in summer.

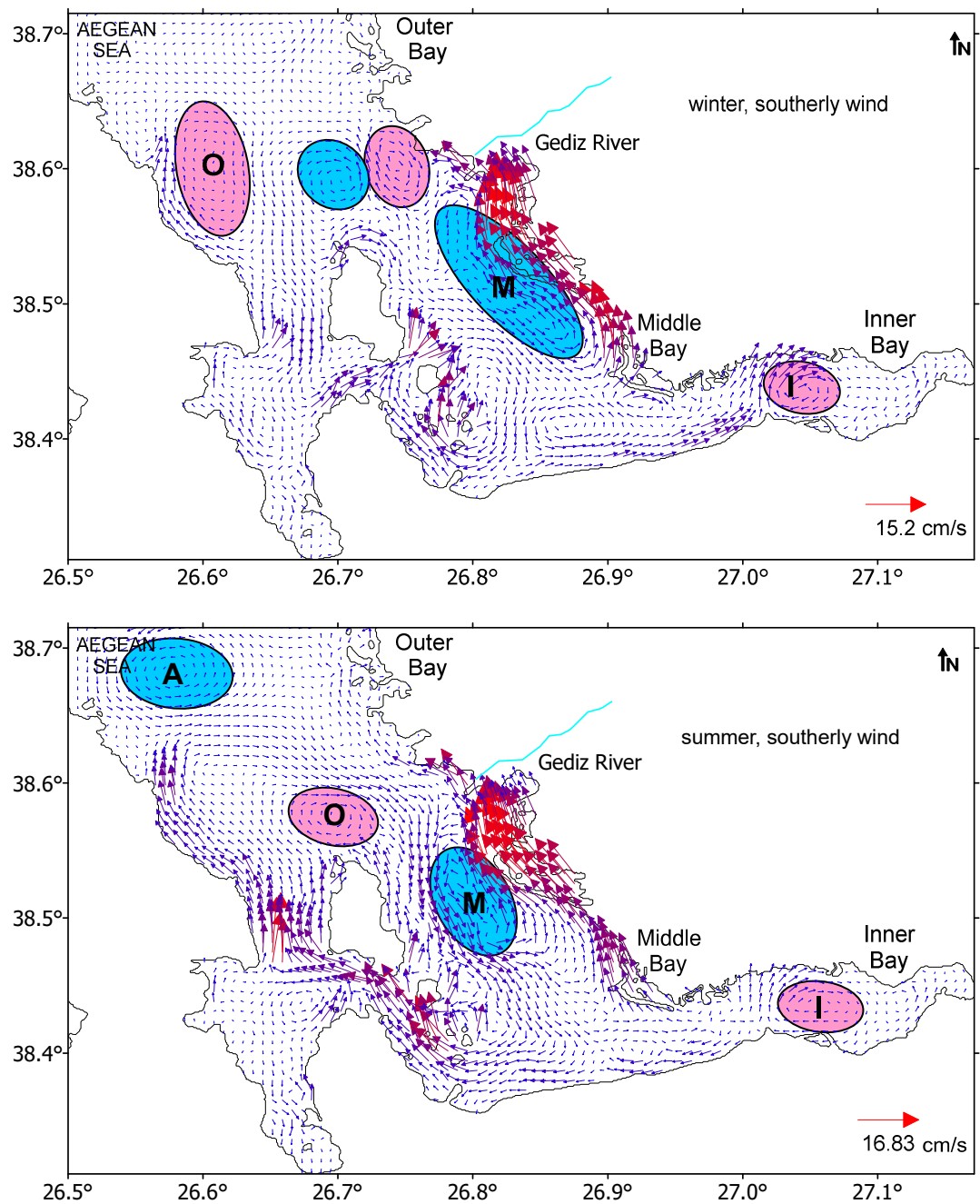

**Fig. 8.** The barotropic current pattern (depth averaged) in case of southerly wind in winter and in summer.


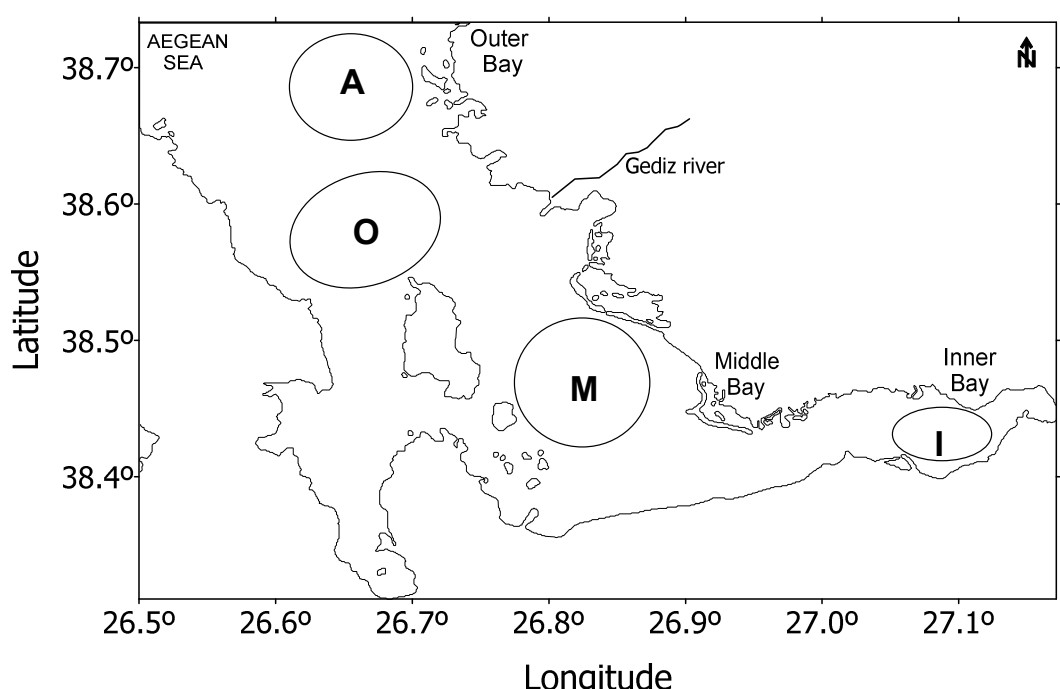

**Fig. 9.** Frequently seen recirculation pattern in the İzmir Bay.