# Peer review of "The dynamics of İzmir Bay under the effects of wind and thermohaline forces"

_Ocean Science, 2017_

## Referee Comment (RC1) · Anonymous Referee #1 · 13 Nov 2017

**General comments**

The manuscript "The dynamics of Izmir Bay under the effects of wind and thermohaline forces" presents results obtained by a general circulation model (GCM) which has been set up up to study circulation of the Izmir Bay/Aegean Sea under varying forcing. The experiments are set up using a) different kinds of artificially generated wind fields and b) the influence of "thermohaline forces" is studied by initialization with temperature and salinity fileds obtained from observational campaigns representative for a summer and winter case study. The Izmir Bay has not received so much perception in the oceanographic community, so I think this study has the potential to present a step forward in Izmir Bay / Aegean Sea research. However, there are several shortcomings in the presentation of results and open questions about experimental strategy and the

general motivation for the study that prevents me from supporting the publication in its present form in Ocean Science.

1) I think but it should be mentioned more prominently the Izmir Bay is relatively unexplored region compared to other regional areas and that available reanalysis data sets and observational data sets to initialize the model are not appropriate for that region. This justifies the more conceptional approach of the current study.

2) What is the motivation to apply this model to this research area. What is the advantage compared to previous approaches to model that region.

3) What is the motivation the study circulation of the Izmir Bay. Why is it important? There are only a few hints in the manuscript distributed at several places.

4) The methods section lack fundamental information about the experimental setup and strategy: How long was the model integrated for each experiment. There are results shown for winter and summer. Are this different experiments or were the summer experiment initialization with circulation regime of the winter experiment? What was the motivation to use the artificial wind fields that were used to force the model. Are there some related to a predominant wind direction over the area. Do they represent a spectrum to cover the main probable directions? When no meteorological information is available this should be explicitly mentioned.

5) The results and discussion presents several a number of different characteristic circulation patterns. The authors should avoid here to remain on an exercise level. Which are the structures and currents that important with respect to the general scientific question of the study? Which structures are important for water mass transfer. Which might be important for sediment transport and coast forming processes (if there are some!). What might be important for biology and ventilation. In its present for its hard to read and one wonders what is the main point here.

6) As I understand, no further mass and energy fluxes (except momentum) at the air
sea boundary were applied. A discussion of how this influences the results would be helpful (especially for the thermohaline case study). Also is the freshwater discharge of the Gediz river accounted for in the simulations? Would be good to know to get insight into the baroclinic behaviour and eddy generation near the river mouth. I Specific comments

line 6: "thermohaline forces". Be more specific, what is the physical force? line 10: two layered"" and "horizontally shared " is very vague and hard to understand line 14: I suggest to reformulate this sentence. line 24: I suggest the term silt sediment or silty deposits rather than silt 25 what is mean by physical characteristics? topography water masses? 28: "silting process" do you mean the continuous filling of the Bay by riverhine sediment loads? I think its not so important for the study whether it is sand, silt clay or muddy material 29 "used to join" - formerly? 35-36 sedimentation accumulates? sediment loss at the bottom due to dissolution, erosion or whatever. Be more precise with what you want to express. 41: when you distinguish different types of water then a few words to characterize the types would be helpful.

43 transport processes through vertical sections. Is there already something known about the renewal time? Mabe in Sayin 2003? then report this here. Later in the MS you present no volume transport calculations to give support for this.

53-66: Several previous studies are mentioned here. The outcome and results for the circulation should be referred a bit more verbose. Which question are still open and which of them do you want to address here with your model. Example: ".Eronat & SayÄśn (2014) studied on the temporal evolution of water characteristic..". But the the reader get not any further information about. If this study is of interest for reader without knowledge on Izmir Bay oceanography you have to give more information.

79 -87: So what is the advantage of this model compared to previous model approaches ? why do you think currents are better represented with this model? Figure
2 shows results for summer and winter for a westerly wind regime. Why westerlies are chosen, how is this wind generated to force the model. What was the reason to chose 5m7s winds at constant rates? Related to observation or theoretical considerations? What is about the model topography, did you use at established data set for this , do you you a flat bottom? Please be more verbose with what you have done to obtain the results. How long were the individual simulation integrated? This is important information.

So is there no heat or water exchange with the atmosphere in all the experiments? Is that right? if so how would this influence the results for the thermohaline circulation. Is the Gediz river water discharge represented in the model? I think this would be important for the discussion of the thermohaline experiments (baroclinic eddy generation etc...) Figure 2: is this the depth averaged salinity and temperature or the first level? what is exactly shown? Please also tell the reader how the CTD measurements are brought onto the model grid? Which is the number of observations that go into the model? Is 30, 300, or 3000. A profound oceanographic analysis of the observations you used would be also a result to present here (if not elsewhere. Is it in agreement with the distinction of the water types you did above?

120: Please indicate this bifurcation during summer in Figure 3. Its hard to see from the description alone

125: " it is almost horizontally homogeneous; but vertically stratified water column changes the behaviour of the current during summer. " Hard understand what is meant here.

145: "The current." what is this certain speed that sets up the this current? 147 Didn't you say previously that you used a constant wind speed of 5m/s. Please give information about how you forced your model. Which was the max. speed?

150ff: Ok you used several kinds of wind directions. Is there something known about what is the main predominating wind direction during the seasons. If so then give this
info and the source.

You describe many different circulation patterns here, but it is not really clear what we can learn from that for the Izmir Bay oceanography. Is there any observational support for this. Or is the existence of the modelled currents any further implications for biology and possible implications sediment transport or so. Otherwise, the article turns of as a more theoretical exercise.

175: "Sometimes...". Is there any explanation for that these features combine sometimes, and sometimes not? Or do we interpret here simply stochastic behaviour? Which is the message the reader could keep in mind here?

The conclusions read very similar to what was mentioned already in the results discussion. Here would be the place for broader implications of the results. What would be the effect of the found recirculation patterns and eddies. How would they act to mix water across the different water types. What would be the implication for biology. Do the results support features from biologists or geologists? Can we draw conclusion for hazardous instances ? The work was apparently supported by the Izmir Marine Research Project. Can we find here some motivation for the study.

OSD

---

## Author Comment (AC1) · 27 Nov 2017

Comment

C1.    I think but it should be mentioned more prominently the Izmir Bay is relatively unexplored region compared to other regional areas and that available reanalysis data sets and observational data sets to initialize the model are not appropriate for that region. This justifies the more conceptual approach of the current study.

Explanation

E1.    Monitoring in İzmir Bay has been initiated seasonally since 1990. Approximately 100 cruises were done up to now. There are sufficient data sets to initialize the model. However, the approach of the study is to analyze the wind driven circulation under different stratification of the Bay. Two representative summer and winter stratifications were chosen in current study. The distribution of stations was important by this cruise selection.

C2.    What is the motivation to apply this model to this research area? What is the advantage compared to previous approaches to model that region?

E2.    Generally, the wind is very important for coastal regions. The deepest part of the İzmir Bay is about 70m. Therefore, lasting strong wind from certain direction generates wind-driven circulation patterns in the Bay. The motivation is to detect these patterns and to know if they change depending on the stratification on the background. The other former model approaches are; i) They study the circulation pattern (snap-shot) in Izmir Bay by using real time wind forces and cruise time stratification. ii) They try to find the circulation regime in the Bay with some restrictions.

C3.    The methods section lacks fundamental information about the experimental setup and strategy: How long was the model integrated for each experiment. There are results shown for winter and summer. Are this different experiments or were the summer experiment initialization with circulation regime of the winter experiment?
What was the motivation to use the artificial wind fields that were used to force the model. Are there some related to a predominant wind direction over the area. Do they represent a spectrum to cover the main probable directions? When no meteorological information is available this should be explicitly mentioned.

E3.    The steady current is achieved by controlling the kinetic energy of the system.  The integration is stopped as soon as the kinetic energy level reaches to a plateau. This information is already in the material and method section. Every experiment has own initialization procedure.
The information about the wind regime of the İzmir Bay environment is given in the section of material and method.
The blowing strong wind in certain direction if it continues long time, approximately more than 12 hours, the current fields in the Bay will go under the influence of the wind and form expected circulation patterns. After run with artificial wind fields, we are able to give information about the circulation patterns to other discipline interested in them. It is not easy to get justification for recirculation patterns from all disciplines. But some remote sensed observations can help.
For example, Figure a shows the TSS (Total Suspended Sediment) distribution in the İzmir Bay in first September 2016 (personal communication, Eronat, 2016, not published). Figure b gives information about the wind intensity and directions before and after date of remote sensed TSS field. As it is noticed that the anti-cyclonic pattern of TSS field has well agreement with the circulation pattern obtained from the model results of northerly wind case.

[Figure]

Figure a) Total Suspended Sediment distribution in the İzmir Bay.

[Figure]

Figure b) The wind intensity and directions (https://www.windguru.cz) before and after date of remote sensed TSS field.

C4.     The results and discussion presents several a number of different characteristic circulation patterns. The authors should avoid here to remain on an exercise level. Which are the structures and currents that important with respect to the general scientific question of the study? Which structures are important for water mass transfer. Which might be important for sediment transport and coast forming processes (if there are some!). What might be important for biology and ventilation. In its present for its hard to read and one wonders what is the main point here.

E4.     We added the importance of different characteristic circulation patterns for biology and related to sediment transport and distribution of some substances in the Bay. On the other hand, the other disciplines do not have enough data resolving the gyres and small features in the Bay. But some very few study (not published) mention about the biological and chemical activity depicting the role of the Middle Gyre. The phytoplankton tends to move up and down in water column depending on the sign of the circulation. Also TSS sinks or comes up to surface as a result of anti-cyclonic or cyclonic movements respectively in the Middle Part of the Bay (E3, Figure a).

C5.     As I understand, no further mass and energy fluxes (except momentum) at the air sea boundary were applied. A discussion of how this influences the results would be helpful (especially for the thermohaline case study). Also is the freshwater discharge of the Gediz

river accounted for in the simulations? Would be good to know to get insight into the baroclinic behavior and eddy generation near the river mouth.

E5.    The reason of the running model in short time is to avoid the meteorological influence which are causing slow change in baroclinic field. It is focused on the fast evolving circulation patterns under the influence of strong lasting wind from certain direction in current study. Sea level can change the circulation pattern. But sea level data is too coarse for İzmir Bay environment. Coarse data brings more difficulties and the obtained results with sea level are far from expected real cases.
In the last years the discharge from Gediz River is reduced drastically because of usage of fresh water for other purposes in the land. The local authorities have built a channel around the Bay to collect the fresh water coming from small streams around since 2000. The leaking fresh water from Gediz and coastal area are occasional and cannot influence whole Bay except in the limited area near coast and near river mouth in rainy days. Therefore, their influences on the baroclinic behavior and eddy generation can be neglected.

Specific Comment
C6.    line 6: "thermohaline forces". Be more specific, what is the physical force?

Explanation
E6.    Thermohaline force here is the force consist of temperature and salinity (density) field differences in space horizontally and vertically along the water column. They turn to geostrophic force together with effect of real topography.

C7.    line 10: two layered"" and "horizontally shared " is very vague and hard to understand

E7.    İzmir Bay which is a coastal shallow area, has vertically two layer in summer. The first layer has high temperature (26° C) and second layer has lower temperature (16° C) and higher salinity. It causes a density difference in the water column. Therefore, the surface currents generally are opposite directions to the lower layer in such a two-layered system. On the other hand, in winter the currents have tendency to flow in one directions. Although existing homogeneous water along the water column in winter, the temperature and salinities in the Outer part of Izmir Bay are always different compared to the temperature and salinities in the Inner part of the Bay. Therefore, we can consider İzmir Bay generally horizontally shared domain in winter.

C8.    line 14: I suggest to reformulate this sentence.

E8.    The sentence "Although the stratification in the bay changes the behaviour of the circulation, the recirculation pattern does not change seasonally, but changes under the influence of wind forcing" is changed. New sentence;
"The lasting strong wind from certain direction generates circulation patterns independent from the seasonal stratification in the Bay".

C9.    line 24: I suggest the term silt sediment or silty deposits rather than silt

E9.    This sentence is quoted from Maddy et al. (2012).

C10.    25 what is mean by physical characteristics? topography water masses?

E10.     The sentence is corrected as: "It can be divided into three areas according to their physical characteristics (containing different water types and bathymetry, etc.): Outer, Middle and Inner Bays, as indicated in Fig. 1".

C11.     28: "silting process" do you mean the continuous filling of the Bay by riverhine sediment loads? I think its not so important for the study whether it is sand, silt clay or muddy material

E11.     It is a priory knowledge that the topography has been changing slowly in years. The topography is important which has influence the currents flowing above it. The model is running with the real bathymetry of the İzmir Bay. It is true that it is not important for the study whether it is sand, silt clay or muddy material.

C12.     29 "used to join" - formerly?

E12.     The sentence has been changed accordingly.

C13.     35-36 sedimentation accumulates? sedimentation may lead to accumulation when the sedimentation rate is larger than sediment loss at the bottom due to dissolution, erosion or whatever. Be more precise with what you want to express.

E13.     It is quoted by Karahan (2002). I think he ment "accumulation" as the sedimentation rate is larger than sediment loss at the bottom.

C14.     41: when you distinguish different types of water then a few words to characterize the types would be helpful.

E14.     Outer Bay water type ASW has a greater volume than the other water types in the bay. Relatively small temporal changes are observed in its temperature and salinity values due to its large volume. Inner Bay water type IBIW is the coldest in winter and its temperature varies from 9.1 °C to 13.9 °C. It has maximum temperature in summer and changes from 24.5 °C to 27.5 °C. IBW is formed in the middle Gyre area, influenced by the Gediz River inflow and by the upwelling and downwelling processes that are mainly driven by southerly and northerly winds, respectively. IBW seems that it is a mixture of IBIW and ASW in winter. But it is very distinguished water type in summer with its higher salinity values varying between 39.6 psu and 39.9 psu.
This information is added to the manuscript.

C15.     43 transport processes through vertical sections. Is there already something known about the renewal time? May be in Sayin 2003? then report this here. Later in the MS you present no volume transport calculations to give support for this.

E15.     The information about renewal time is added as:
The renewal time considering the water exchange through the vertical section between İzmir Bay and Aegean Sea is found by 46 and 29 days for the winter and summer case, respectively (Sayın, 2003).

C16.     53-66: Several previous studies are mentioned here. The outcome and results for the circulation should be referred a bit more verbose. Which question are still open and which of them do you want to address here with your model. Example: ".Eronat & Sayin (2014) studied on the temporal evolution of water characteristic..". But the reader get not any further

information about. If this study is of interest for reader without knowledge on Izmir Bay oceanography you have to give more information.

E16.     Some information is added.

C17.     79 -87: So what is the advantage of this model compared to previous model approaches? why do you think currents are better represented with this model?

E17.     All model mentioned in the manuscript use the Navier-Stokes equations and are able to model İzmir Bay. But they have different approaches. These approaches are explained before in the explanation E2.

C18.     Figure 2 shows results for summer and winter for a westerly wind regime. Why westerlies are chosen, how is this wind generated to force the model.

E18.     Figure 2 shows the winter and summer temperature and salinity fields in the Bay. It is not related to wind-driven circulation.

C19.     What was the reason to chose 5m/s winds at constant rates? Related to observation or theoretical considerations?

E19.     First we integrated the model until the steady current is achieved by controlling the kinetic energy of the system.  Afterwards it is seen that 5 m/s wind intensity was sufficient for resolving eddy circulation.

C20.     What is about the model topography, did you use at established data set for this, do you a flat bottom? Please be more verbose with what you have done to obtain the results.

E20.     The model experiments are conducted using real topography. This information is added to method section.

C21.     How long were the individual simulation integrated? This is important information.

E21.     This integration procedure has been already explained by E2.

C22.     So is there no heat or water exchange with the atmosphere in all the experiments? Is that right? if so how would this influence the results for the thermohaline circulation. Is the Gediz river water discharge represented in the model? I think this would be important for the discussion of the thermohaline experiments (baroclinic eddy generation etc.)

E22.     The necessary explanations to the arising point has been done already in E5.

C23.     Figure 2: is this the depth averaged salinity and temperature or the first level? what is exactly shown? Please also tell the reader how the CTD measurements are brought onto the model grid?
E23.     Figure 2 shows the winter and summer, temperature and salinity fields of 5 m that are prepared to give to the model as temperature and salinity distribution of the first level. Surfer Program has been used to distribute the temperature and salinity values to the model grids. The grids are prepared for the Model using a Fortran program creating a file to be read by the Model.

C24.     Which is the number of observations that go into the model? Is 30, 300, or 3000. A profound oceanographic analysis of the observations you used would be also a result to

present here (if not elsewhere. Is it in agreement with the distinction of the water types you did above?

E24.    Temperature and salinity values of approximately 40 CTD casts are used for each model experiment. The reason to choose these cruise data is that the data is representable for the physical oceanography of İzmir Bay indicating the water types already explained by Sayin et al., 2006.

C25.    120: Please indicate this bifurcation during summer in Figure 3. Its hard to see from the description alone

E25.    This Figure is corrected.

C26.    125: " it is almost horizontally homogeneous; but vertically stratified water column changes the behaviour of the current during summer. " Hard understand what is meant here.

E26.    The point is cleared in the Explanation E7.

C27.    145: "The current.." what is this certain speed that sets up the this current?

E27.    The numerical experiment was conducted to show the development of circulation in the Inner Bay by increasing the wind intensity from zero to 5 m/s. The current, not only in the Inner Bay, but also in the other regions of the Bay starts to set up after a certain wind speed is exceeded. The current is very weak in the Inner Bay without the existence of wind force. The currents get stronger with increasing wind speed. Recirculation patterns which exist in the Middle Bay become well-developed after the increase of wind intensity above approximately 2.5 m/s and are observable both in the barotropic field and in the certain layers.

C28.    147: Didn't you say previously that you used a constant wind speed of 5m/s. Please give information about how you forced your model. Which was the max. speed?

E28.    Explanation E27 is also applicable for the current Comment C28.

C29.    150ff: Ok you used several kinds of wind directions. Is there something known about what is the main predominating wind direction during the seasons. If so then give this info and the source.

E29.    The predominant wind condition of Izmir environment is added to the Method section and shown in the Figure **c** below.

[Figure]

Figure c. Wind direction distribution from 1985 up to now (https://www.windfinder.com).

C30.    You describe many different circulation patterns here, but it is not really clear what we can learn from that for the Izmir Bay oceanography. Is there any observational support for this. Or is the existence of the modelled currents any further implications for biology and possible implications sediment transport or so. Otherwise, the article turns of as a more theoretical exercise.

E30.    The required explanation has been given already in E3.

C31.    175: "Sometimes...". Is there any explanation for that these features combine sometimes, and sometimes not? Or do we interpret here simply stochastic behaviour? Which is the message the reader could keep in mind here?

E31.    If the sign of **M** and **O** are same and **M** is very near to Outer Bay, these features combine each other depending on the direction of wind. This information is added to the manuscript.

C32.    The conclusions read very similar to what was mentioned already in the results discussion. Here would be the place for broader implications of the results. What would be the effect of the found recirculation patterns and eddies. How would they act to mix water across the different water types? What would be the implication for biology? Do the results support features from biologists or geologists? Can we draw conclusion for hazardous instances? The work was apparently supported by the Izmir Marine Research Project. Can we find here some motivation for the study?

E32.    E3 and E4 explain why the study do not cover biology and other discipline implications. Without any information (published document) from the biologists and geologists it is hard to combine all disciplines to make analysis together. The chemical oceanographer, biologist and physical oceanographer make research together in Izmir Marine Research Project. But a synthesis could not be done because of disciplines having different monitoring purposes. I hope one (group) can get initiative to realize this very valuable issue. Thank the reviewer that remember us this important point.

---

## Author Comment (AC2) · 27 Nov 2017

[revised manuscript text omitted]

**Figures**

---

## Referee Comment (RC2) · Anonymous Referee #2 · 5 Dec 2017

The manuscript "The dynamics of Izmir Bay under the effects of wind and thermohaline forces" studies the circulation of Izmir bay using a 3-D general circulation numerical model. A z-level free surface version of the Princeton model is used as the ocean model. The model is initialized using selected winter and summer hydrological cruise CTD data. The main aim of the paper is to study the effect of two forcing, i. e. wind and thermohaline, on the circulation of Izmir bay. Therefore, two sets of numerical experiments are carried out. In the first set of runs, the model is initialized using the CTD data and run without any other external forcing, including wind and heat fluxes. The second set of runs is similar to the first set, except that wind forcing is included. The wind forcing includes artificial constant wind from four main directions. The model is run until a steady current is achieved, i.e. when the kinetic energy level reaches a

plateau. The circulation patterns are then studied under different forcing.

Considering the availability of long-term observational data in the Izmir bay and the scarcity of studies using numerical modeling for this region, this research has the potential to enhance the understanding of circulation of Izmir bay. However, I think that the paper needs major revisions as suggested:

1. In general some parts of the paper are difficult to follow and could be more coherent (e.g. the introduction). In my opinion, if some sentences (as suggested in the last section of this review) are rephrased, it be beneficial for a better understanding of the research. Considering that the previous papers by the same authors (e.g. Sayin 2003 & Sayin et al 2006) have a good structure and coherence, I think it is possible and worthwhile to revise this manuscript to have a more seamless structure.

2. Could you please present the reason for using an artificial wind forcing? Is it not possible to run the model with realistic data from synoptic stations or re-forecast atmospheric models? The application of using non-realistic forcing has a major drawback that the model cannot be validated against observational data. My understanding is that model validation is an important part of numerical modeling. Therefore, I would suggest that, if possible, the model be run with realistic forcing and after validation, be used for studies under different forcing such as artificial winds.

3. In my opinion, the result section can benefit from more explanation on the physical reason behind the occurrence of predicted patterns. The use of artificial wind in this research could be compensated by describing the physical reason behind the formation of different patterns. Although some explanation is given in this section, a more indepth study is constructive.

Regarding each section I have these suggestions:

Abstract:

A.1 I think it is beneficial to mention that artificial wind is used to force the model so
that the reader does not expect a realistic model setup.

Introduction:

I.1. Although the introduction points out the importance of Izmir bay, in my opinion this section can benefit from a more coherent structure. Also, this section provides some extra information which I think is better to omit to increase focus on the main aim of the paper. For example, the reason for extra-shallow regions in the inner bay is not directly relevant to the study and I suggest removing these lines (28-32).

I.2. I would also suggest that more details on findings of previous research be given and the merit of the present research compared to previous studies be discussed. For example, in line 71 it is mentioned that Sayin (2003) has investigated the physical features based on modeling studies but no further explanation is given on the findings of his research. Similarly in line 66 the findings of Saner is not given.

1.3. In line 31 it is mentioned that elevation gradient maintained in the sea level affects the circulation. First, it is beneficial to include the source of this statement. Second, does this statement not contradict the statement given in lines 116-119?

Materials and methods: M.1. How are the vertical profiles of temperature and salinity in Izmir bay? Since a set of experiments focus on the effect of thermohaline forcing on the circulation, it would be helpful to add an explanation on how the profiles change in each season.

M.2. It is mentioned that at the open boundary observed temperature and salinity is applied. What is the frequency of these data? Are these data constant in time?

M.3. It is mentioned that the model is run until an equilibrium is reached. How long does it take for this steady current to occur? I think adding this explanation to the paper helps in understanding the nature of these forcing.

M.4. According to the supplementary explanations given as an answer to the first reviewer's comments, the wind is increased from 0 to 5 m/s in the experiments. Is that
right? If so, in order to avoid confusion, I suggest to correct line 102 to include this explanation.

M.5. It is mentioned that the wind intensity is chosen to be 5 m/s but there is no reason as to why this speed is used. Is this speed chosen according to observed wind in Izmir bay? How does the wind change seasonally? This may be important in analyzing the effect of wind and stratification on the results. I think these questions can be answered by including the wind-rose in the paper and adding some explanation about the dominant wind and its intensity, if possible for summer and winter.

Results:

R1. As mentioned before, I think it is beneficial that a more in-depth explanation be given to why the described patterns occur rather than adhering to describing the patterns. For example in line 140, it is mentioned that the M pattern changes sign from winter to summer. Is it possible to give an explanation to why this happens?

R2. Are the current fields depicted in the figures depth-averaged fields? If so this should be indicated in the figures captions.

Conclusion:

C.1. Regarding the conclusion, I also have the opinion that this part is only a repetition of the results section.

Regarding the change in sentences I suggest to rephrase these lines:

Line 10: although I understand the meaning of this sentence by looking at the answer to the first reviewer comments, I think this sentence is still not clear for the reader and should be rephrased.

Line 14: The lasting strong wind from certain direction: be more specific, what is meant by certain directions.

Line 18: Outer and Inner Bay have also certain wind driven recirculation patterns:

OSD
Again what is meant by certain? I think vague statements should be avoided in the abstract.

Line 63: One can increase the number of examples : rephrase

Line 131: turns to the direction to: rephrase

Line 133and other branch complete the cyclonic circulation in basin wide: rephrase

Line 135: Instead of being vertically homogeneous, it is almost horizontally homogeneous; but vertically stratified water column changes the behaviour of the current during summer: rephrase

Line 138 - 140 : rephrase

- Line 161 : the certain layers : what does certain mean here?
- Line 163 : It is preferred to explain the current system : rephrase

I also suggest to change these sentences as follows:

- Line 3: wind is the most important driving-force (instead of driven-force)
- Line 14: Change "Lasting strong wind" to "Strong consistent wind"
- Line 33: The water input through Gediz River is relatively low (instead of small)
- Line 77: In (the) present study (add article)

Line 80: omit "the information"

- Line 94: reaches a plateau (omit to)
- Line 110: omit "which were done"
- Line 110: "deals with" : change to "focuses on"
- Line 135: "comparison to" : change to "compared to"

---

## Author Comment (AC3) · 18 Dec 2017

The manuscript "The dynamics of Izmir Bay under the effects of wind and thermohaline forces" studies the circulation of Izmir bay using a 3-D general circulation numerical model. A z-level free surface version of the Princeton model is used as the ocean model. The model is initialized using selected winter and summer hydrological cruise CTD data. The main aim of the paper is to study the effect of two forcing, i. e. wind and thermohaline, on the circulation of Izmir bay. Therefore, two sets of numerical experiments are carried out. In the first set of runs, the model is initialized using the CTD data and run without any other external forcing, including wind and heat fluxes. The second set of runs is similar to the first set, except that wind forcing is included. The wind forcing includes artificial constant wind from four main directions. The model is run until a steady current is achieved, i.e. when the kinetic energy level reaches a plateau. The circulation patterns are then studied under different forcing. Considering the availability of long-term observational data in the Izmir bay and the scarcity of studies using numerical modeling for this region, this research has the potential to enhance the understanding of circulation of Izmir bay. However, I think that the paper needs major revisions as suggested:

1. In general some parts of the paper are difficult to follow and could be more coherent (e.g. the introduction). In my opinion, if some sentences (as suggested in the last section of this review) are rephrased, it be beneficial for a better understanding of the research. Considering that the previous papers by the same authors (e.g. Sayin 2003 & Sayin et al 2006) have a good structure and coherence, I think it is possible and worthwhile to revise this manuscript to have a more seamless structure.
E.1. The introduction section is rewritten.

2. Could you please present the reason for using an artificial wind forcing? Is it not possible to run the model with realistic data from synoptic stations or re-forecast atmospheric models? The application of using non-realistic forcing has a major drawback that the model cannot be validated against observational data. My understanding is that model validation is an important part of numerical modeling. Therefore, I would suggest that, if possible, the model be run with realistic forcing and after validation, be used for studies under different forcing such as artificial winds.
E.2. The model has been using for a long time in the Institute of Marine Science and Technology. The validation with observations has been carried out and first results have been obtained by Sayın (2003) and Sayin et al 2006. The comparison with the other models in institute (a primitive equation model of the Harvard Ocean Model and FVCOM (the Unstructured Grid Finite Volume Community Ocean Model)) are done and achieved good agreement. The model was previously applied to the Baltic Sea and the straits between Baltic and North Sea (Sayin and Krauss, 1996) justifying that this model can be used also for small seas as well as for straits and channels. Sayin et al 2006 has run the model with realistic forcing. In the present study, our motivation is to understand the behavior of the current field under blowing strong wind from four main directions. If the wind blows long time from the same direction continuously, approximately more than 12 hours, the current fields in the Bay will go under the influence of wind and form wind-driven circulation. It will help to the biologist, chemist and other researchers and they will have detailed pattern about the current field, if they know from which direction the wind blows.

3. In my opinion, the result section can benefit from more explanation on the physical reason behind the occurrence of predicted patterns. The use of artificial wind in this research could be compensated by describing the physical reason behind the formation of different patterns. Although some explanation is given in this section, a more indepth study is constructive.
Persistent wind changes thermohaline circulation and the water is immediately under the influence of wind force in shallow coastal area. Coastal jets are produced along both coasts in the wind direction and a slow return flow compensates this transport in the central area of the basin as explained by Krauss and

Brügge (1991). For example, in case of southerly wind it could be seen the establishing coastal jets and developing cyclonic gyre in the middle of the İzmir Bay is the place for the formation of the dense water.

Regarding each section I have these suggestions:
Abstract:
A.1 I think it is beneficial to mention that artificial wind is used to force the model so that the reader does not expect a realistic model setup.
Usage of artificial wind is added to the abstract.

Introduction:
I.1. Although the introduction points out the importance of Izmir bay, in my opinion this section can benefit from a more coherent structure. Also, this section provides some extra information which I think is better to omit to increase focus on the main aim of the paper. For example, the reason for extra-shallow regions in the inner bay is not directly relevant to the study and I suggest removing these lines (28-32).
The lines are removed and the introduction section is rewritten.

I.2. I would also suggest that more details on findings of previous research be given and the merit of the present research compared to previous studies be discussed. For example, in line 71 it is mentioned that Sayin (2003) has investigated the physical features based on modeling studies but no further explanation is given on the findings of his research. Similarly in line 66 the findings of Saner is not given.
The introduction section is rewritten taking consideration of the reviewer comment. We appreciate the priori modelling efforts. Any further explanation about Saner (1994) and Saner (2005) model studies are added, because his model results are relevant for the engineering point of view. He has compared his two model approaches mathematically.

1.3. In line 31 it is mentioned that elevation gradient maintained in the sea level affects the circulation. First, it is beneficial to include the source of this statement. Second, does this statement not contradict the statement given in lines 116-119?
Although surface elevation gradient generally is an important driving mechanism for the forming barotropic currents. Because of not having appropriate sea level data, the model experiments are conducted without adding sea level in the model. This issue can be a future research effort for the İzmir Bay modelling studies.

Materials and methods:
M.1. How are the vertical profiles of temperature and salinity in Izmir bay? Since a set of experiments focus on the effect of thermohaline forcing on the circulation, it would be helpful to add an explanation on how the profiles change in each season.
The figures are added.

M.2. It is mentioned that at the open boundary observed temperature and salinity is applied. What is the frequency of these data? Are these data constant in time?
Observed temperature and salinity values are prescribed at the boundary and relaxed during rest model integration time. This information is added to the text.

M.3. It is mentioned that the model is run until an equilibrium is reached. How long does it take for this steady current to occur? I think adding this explanation to the paper helps in understanding the nature of these forcing.
It is variable for every run. But it takes approximately three days. The background stratification remains not changed from its original prescribed form because of the equilibrium is succeeded in a short time. This is the importance of wind-driven scenarios with constant wind intensity. Model will not be in a steady state if we use actual wind field to run the model.

M.4. According to the supplementary explanations given as an answer to the first reviewer's comments, the wind is increased from 0 to 5 m/s in the experiments. Is that right? If so, in order to avoid confusion, I suggest to correct line 102 to include this explanation.

The suggested information is added to the text.

M.5. It is mentioned that the wind intensity is chosen to be 5 m/s but there is no reason as to why this speed is used. Is this speed chosen according to observed wind in Izmir bay? How does the wind change seasonally? This may be important in analyzing the effect of wind and stratification on the results. I think these questions can be answered by including the wind-rose in the paper and adding some explanation about the dominant wind and its intensity, if possible for summer and winter.

The dominant wind and its intensity for İzmir Bay environment is demonstrated using a wind-arrow graphic in the materials and methods section. Wind direction and average wind intensity from 2000 up to now monthly and yearly for İzmir Bay environment shows that the wind from north is predominant direction and the average wind speed is 5 m/s. Therefore 5 m/s wind intensity is chosen to simulate persistent wind condition.

Results:

R1. As mentioned before, I think it is beneficial that a more in-depth explanation be given to why the described patterns occur rather than adhering to describing the patterns. For example, in line 140, it is mentioned that the M pattern changes sign from winter to summer. Is it possible to give an explanation to why this happens?

The patterns, that are seen as a result of density-driven model experiments, can vary in time and space depending on the background distribution of temperature and salinity in the İzmir Bay model domain. So it is not correct to conclude that the pattern change sign from winter to summer. We reorganized the abstract section and removed the statement related the patterns for density-driven case. But result section is changed slightly by including information about the general distribution of density field in the Bay.

R2. Are the current fields depicted in the figures depth-averaged fields? If so this should be indicated in the figures captions.

The figure captions are rearranged indicating the current fields depicted in the figures are depth-averaged fields.

Conclusion:

C.1. Regarding the conclusion, I also have the opinion that this part is only a repetition of the results section.

The importance of pattern **M** is added to the conclusion section:

The cyclonic middle gyre M is important for İzmir Bay environment from two points. First is related to the dense water formation. The densest water (IBW) forms in the Middle Bay as a result of winter convection enhanced with cyclonic circulation in winter season. It causes a dense water cascading from İzmir Bay to Aegean Sea. Second is important from the biological point of view, forming upwelling brings nutrients rich water to the surface.

Regarding the change in sentences I suggest to rephrase these lines:

Line 10: although I understand the meaning of this sentence by looking at the answer to the first reviewer comments, I think this sentence is still not clear for the reader and should be rephrased.

The sentence is changed to "Instead two-layer stratification during summer, a homogeneous water column exists in winter".

Line 14: The lasting strong wind from certain direction: be more specific, what is meant by certain directions.

The sentence is rephrased.

Line 18: Outer and Inner Bay have also certain wind driven recirculation patterns: Again what is meant by certain? I think vague statements should be avoided in the abstract.

"and Inner Bay have also certain wind driven recirculation patterns" is replaced by "and Inner Bay have also expected wind driven recirculation patterns"

Line 63: One can increase the number of examples: rephrase

The sentence "One can increase the number of examples in which the currents and background-forming horizontal and vertical stratification are crucial for marine environments" is removed and the paragraph is rephrased.

Line 131: turns to the direction to: rephrase
The paragraph is rewritten and rephrased replacing "turns to the direction to" with "turns towards".

Line 133and other branch complete the cyclonic circulation in basin wide: rephrase
The paragraph is rewritten and rephrased replacing "and other branch complete the cyclonic circulation in basin wide" with "and other branch combines with the strong current at the east coast making a big cyclonic circulation in the middle area".

Line 135: Instead of being vertically homogeneous, it is almost horizontally homogeneous; but vertically stratified water column changes the behaviour of the current during summer: rephrase
The paragraph is reorganized taking attention reviewer suggestion.

Line 138 – 140 : rephrase
It is rephrased.

Line 161 : the certain layers : what does certain mean here?
It is rephrased writing "vertical" instead of "certain"

Line 163 : It is preferred to explain the current system : rephrase
It is rephrased as "The current system is explained giving emphasis only to the recirculation patterns forming in the Bay"

I also suggest to change these sentences as follows:
Line 3: wind is the most important driving-force (instead of driven-force)
It is replaced with "driving-force"

Line 14: Change "Lasting strong wind" to "Strong consistent wind"
"Lasting strong wind" is replaced with "Strong consistent wind"

Line 33: The water input through Gediz River is relatively low (instead of small)
"small" is replaced by "low"

Line 77: In (the) present study (add article)
(the) is added

Line 80: omit "the information"
"the information" is omitted.

Line 94: reaches a plateau (omit to)
"to" is omitted.

Line 110: omit "which were done"
"which were done" is omitted

Line 110: "deals with": change to "focuses on"
"deals with" is changed to "focuses on"

Line 135: "comparison to": change to "compared to"
"comparison to" is changed to "compared to"

---

## Author Comment (AC4) · 18 Dec 2017

[revised manuscript text omitted]

---

## Referee Report (RR1)

**Comments on the revised manuscript:**
**"The dynamics of İzmir Bay under the effects of wind and thermohaline forces"**
**by Erdem Sayın and Canan Eronat**

The revised manuscript is an improvement compared to the original version. The introduction and aim of the study is now clear and more detailed description is given in the results and discussion section. The research will contribute to a better understanding of the İzmir Bay dynamics and therefore I suggest that the manuscript be accepted with some minor technical (grammatical) corrections. In my opinion, the manuscript can benefit from proofreading for a better understanding. Some examples of improvement are given below.

Examples for improvement:
Line 10 : replace "Instead a two layer" with "Instead of a two-layer "
Line 14: replace "independent from" with "independent of"

Examples of sentences that could be rephrased:
Line 44 : The renewal time, which is important from the biological and chemical
point of view, considering the water exchange through the vertical section between İzmir Bay
and Aegean Sea is found by 46 and 29 days for the winter and summer case, respectively.
Line 118: wind intensity from 2000 up to now monthly and yearly for
Line 121: in the model experiments testing which wind intensity
enough to simulate the strong wind condition
Line 126: prescribed at the boundary and relaxed during rest model integration time
Line 135: the other group with the wind-driven circulation.
Line 167: Comparison winter patterns with the patterns
Line 232: has an anticyclonic character in case northerly and westerly winds
Line 251: The prognostic modelling approach can be a future challenge for the modelling of
İzmir Bay with adding more meteorological information inside.

---

## Author Response (AR2)

Explanation to
**Referee #2**

I think the manuscript can be published with minor corrections.
The introduction is now more concise and gives a good overview over the purpose of the study and previous work on that. Likewise, important information in the methods section has been added and it is now possible to more easily understand the modeling approach and experimental strategy. Drawbacks of the study are clearly indicated and are discussed under the framework of the studies goals and available current knowledge of the Aegean Sea and Izmir Bays oceanography..
The discussion and the results are now better understandable. Improvements and addition of figures likewise facilitate the understanding of the studies and one gets now insight what the findings of this study are valid for.

I have only some minor comments:

Abstract. I recommend to remove the last sentence.
The last sentence is removed from the abstract.

line 52: rephrase this sentence. Maybe:
"Using a three dimensional finite difference model the author demonstrated that the shallowing..."
The sentence is rephrased as "Using a three dimensional finite difference model the author demonstrated that the shallowing which is in the middle bay entrance has effect on the water input and output of outer bay".

line 70: put 2006 into paranthesis (2006)
2006 is written in parenthesis.

line 73 rephrase like e.g.:
There is a demand by other disciplines such as biologists chem8ists etc to get detailed information about dominant circulation patterns under cahracterisyc wind fields.
The sentence is rephrased as suggested above.

line 95 rephrase:
"Since 1996 regular measeurements have been conducted..."
The sentence is rephrased as suggested above.

line 109 in OUR institute
"in the Institute of Marine Science and Technology" is replaced with "in our institute".

line 112 Sayin and Krauss, 1996 is lacking in the references.
Sayin and Krauss, 1996 is added to the reference list.

line 164 is MORE complicated
"more" is added before "complicated".

line 194 UNDER westerly conditions
"in" is replaced with "under".

line 240: How is winter convection produced when meteorological forcing is absent?
This paragraph explains the importance of the Middle Gyre. The cyclonic circulation takes place mainly as a result of southerly wind that is simulated also in the model. On the other hand, the temperature data obtained from monitoring studies shows us how strong the winter convection took place in cold seasons.

line 241 replace "Second" by "The latter"
"Second" is replaced by "The latter".